# Educational considerations for health professionals to effectively work with clients with complex regional pain syndrome

**Colleen Johnston-Devin** [ID][1]*, **Florin Oprescu**[2]◉, **Marianne Wallis**[3]◉, **Marion Gray**[4]◉

1 School of Nursing, Midwifery and Social Sciences, Central Queensland University, Brisbane, Queensland, Australia, 2 School of Health and Sport Science, University of the Sunshine Coast, Sippy Downs, Queensland, Australia, 3 Faculty of Health, Southern Cross University, Bilinga, Queensland, Australia, 4 School of Health and Wellbeing, University of Southern Queensland, Ipswich, Queensland, Australia

◉ These authors contributed equally to this work.
* c.johnston-devin@cqu.edu.au

**Data Availability Statement:** All relevant data are within the manuscript and available in the thesis: Battling Complex Regional Pain Syndrome (CRPS): A Phenomenological Study https://doi.org/10. 25907/00041.

## Abstract

### Introduction

People living with complex regional pain syndrome (CRPS), a rare chronic pain disorder, must become experts in their own self-management. Listening to the voice of the patient is often advocated in the pain literature. However, the patient's option is rarely asked for or considered by clinicians, even when they live with a condition that health professionals have rarely heard of.

### Purpose

To explore what people living with complex regional pain syndrome (CRPS) think health professionals should know about their condition to provide appropriate care.

### Design

A heuristic, hermeneutic phenomenological study was conducted asking people about their experiences living with CRPS. This paper reports on the findings of an additional question asked of all participants.

### Participants

Seventeen people living with complex regional pain syndrome were interviewed.

### Findings

Overwhelmingly, participants felt that health professionals do not know enough about CRPS, or chronic pain and believe their health outcomes are affected by this lack of knowledge. Sub-themes identified were don't touch unless I say it is okay; be patient with the patient/ it is important to develop a relationship; educate yourself and educate the patient; choose your words carefully and refer to others as needed. An additional theme, it is very hard to describe CRPS was also identified.

**Funding:** The authors received no specific funding for this work.

**Competing interests:** The authors have declared that no competing interests exist.

## Conclusions

Including patients as a member of the healthcare team is recommended to help people take control and self-manage their pain. For true patient centered care to be achieved, health professionals must accept and respect patients' descriptions of pain and their pain experience. This may require additional health professional education at both undergraduate and postgraduate levels in pain and communication to increase their bedside manner and therapeutic communication to deliver care in partnership with the patient.

## Introduction

Complex Regional Pain Syndrome (CRPS) is considered to be an immunoneurological disorder [1]. Affecting females more than males, recent research suggests that prior infection or chronic inflammation increases the risk of CRPS developing but the cause has not yet been determined [1]. CRPS most often occurs after fractures and is more common in upper extremities, with approximately 60% of CRPS originating in the arm and 40% in the leg [2, 3]. Although intravenous cannula insertion, injections, pregnancy, mild trauma, minor surgery, sprains, immobilisation and stroke or myocardial infarction are known causes, CRPS can also occur spontaneously [4–6]. The average age of incidence ranges from 50 to 70 years of age, although young children have also been diagnosed [5, 7]. There are a variety of presentations; some patients report symptoms such as hyperalgesia, oedema, temperature and colour changes, allodynia, increased hair growth and abnormal sweating, but the common symptom is pain disproportionate to the precipitating event [8].

Despite clear diagnostic guidelines (called the Budapest criteria), diagnosis of CRPS can take years [9]. The consequences of having CRPS can be debilitating and people with CRPS generally have a poorer quality of life, and a higher risk of suicide than people with other chronic pain conditions [10]. As with other persistent pain conditions, there is a high monetary cost to living with CRPS. The global economic burden of pain which lasts for longer than three months is counted in billions of dollars and in the United States, chronic pain costs more than the combined costs of cancer, heart disease and diabetes [11]. In Australia, these costs are comprised of reduction in quality of life costs, productivity losses, health system costs (such as hospitalisation and pharmaceuticals) and other costs such as modifications, aids and informal costs of care [12]. Considering the high incidence, damaging psychological effects and high monetary costs, it is important that chronic pain treatment and management are addressed appropriately. Complex regional pain syndrome (CRPS) has been reported to be the most painful condition experienced, but, being rare, it is mostly not known within the health professional community and is even less known within the general public [5, 13]. When diagnosed with a rare condition, lack of knowledge about the disorder can negatively impact the level of care provided and hence, the patient can suffer, further increasing both personal and financial costs.

While researchers worldwide attempt to understand the pathophysiology, diagnosis and treatment of CRPS, less attention has been given to understanding the patients' experience of living with the condition. Much can be learned from listening to the voice of patients and this information can be an invaluable tool for current and future health professionals to improve knowledge and provide more effective care. The notions of listening to the patients' voice and patient-centred care are advocated in the chronic pain literature and calls are made by various authors to put the patient at the centre of care and education [14, 15]. Patients, however, are

rarely asked for their opinion. This paper reports the findings of a specific question asked of participants in a study examining the lived experience of CRPS. Thus, the research question underpinning this aspect of the larger study was: What do people who live with complex regional pain syndrome want health professionals to know about their condition?

## Methods

### Design

A heuristic, hermeneutic phenomenological study was undertaken to examine the lived experience of CRPS and further, to ascertain what people living with CRPS thought their health professionals should know to improve their health outcomes. The hermeneutic phenomenological design chosen was informed by van Manen [16]. Acknowledging that the "(phenomenological) facts of lived experience are already meaningfully (hermeneutically) experienced" [16 p181], such research aims to describe, understand and interpret the experiences of the participants [17]. This approach to phenomenology was chosen due to it using descriptions of lived experiences as data to describe the essence of a phenomenon [16]. This article reports on the data related to what health professionals should know about CRPS. The lived experience of living with CRPS is presented elsewhere [18].

**Ethical considerations.** Ethics approval was obtained from the University of the Sunshine Coast Human Research Ethics Committee (Approval number S/13/577) before any research activities were undertaken, and each participant provided written informed consent. To maintain participant anonymity, before each interview, participants chose their pseudonyms to be used throughout the study and in publications.

### Sample/Participants

People living with CRPS were sought to provide their experiences living with this condition and offer their advice to health professionals. While international guidelines support early diagnosis to improve patient outcomes, delayed diagnosis and treatment are widely reported [19, 20]. A purposive sample of 17 people living with CRPS in Australia was recruited from online forums initially, then, internet-based support groups worldwide were approached. Contact was made initially with the administrators of CRPS support groups found on Facebook and Google. After gaining administration approval, information about the study was posted asking for people interested in participating to contact the first author. Every person who responded was sent the Participant Information Sheet and was given the opportunity to ask questions.

The inclusion criteria were the ability to provide informed consent and be interviewed in English during the data collection time frame which was between 2016 and 2017. All participants must have been diagnosed formally with CRPS, but proof of diagnosis was not required. This was consistent with previous studies such as Lewis et al. [21], Packham et al. [22] and Ten Brink et al. [23]. Phenomenology does not strive for data saturation and so to gain as much information as possible, every person who fit the criteria was interviewed [24]. This resulted in 17 interviews meeting the inclusion criteria and being included in the study. These participants lived in Australia, England, Wales, Singapore, and the United States of America.

The age range of the 17 people interviewed for this study was from 22 to 65 years (mean = 44 years). Male to female demographics in this research were in keeping with the one male to three to four females described in the literature and included 14 women and three men [5, 7]. They had a mean lived time with CRPS symptoms of seven years and the range was from four months to 18 years. The mean time from symptom onset to formal diagnosis was 2.65 years with the longest time to a formal diagnosis being nine years and the shortest being three weeks as seen in Table 1. The table also provides an indication of how many different

**Table 1. Patient participant demographics.**

| Chosen name | Age | Time with CRPS | Time to formal diagnosis | Affected body part | Surgeon* | Pain Specialist | Pain clinic ** | Allied Health |
|---|---|---|---|---|---|---|---|---|
| Karen | 55 | 7 years | 3 years | Foot | Yes | Yes | No | Yes |
| Laura | 29 | 15 years | 9 years | Leg | Yes | Yes | No | Yes |
| Rosemary | 64 | 8 years | 5 years | Foot | Yes | Yes | Yes | Yes |
| Colleen | 48 | 4.5 years | 3 weeks | Hand | Yes | Yes | Yes | Yes |
| Sharon | 46 | 5 years | 3 years | Hand | No | Yes | Yes | Yes |
| Fred | 41 | 11 years | 2½ years | Arm | Yes | Yes | Yes | Yes |
| Martin | 32 | 5 years | 18 months | Foot | Yes | Yes | Yes | Yes |
| Emma | 45 | 1 year | 3–4 weeks | Hand | Yes | No | No | Yes |
| Jackie | 55 | 4 months | 3 months | Hand | Yes | Yes | No | Yes |
| Hannah | 22 | 8 years | 2 years | Hand | No | Yes | No | Yes |
| Alice | 30 | 5 years | 9 months | Leg | No | Yes | No | Yes |
| Dianne | 50 | 4.5 years | 3 years | Shoulder | Yes | Yes | Yes | Yes |
| Jasmine | 25 | 13 years | 12 months | Ankle | Yes | Yes | Yes | Yes |
| Sarah | 45 | 16 months | 13 months | Knee | Yes | Yes | Yes | Yes |
| Mel | 41 | 9 years | 4.5 years | Shoulder | Yes | Yes | Yes | Yes |
| Carolyn | 65 | 18 years | 3 years | Arm / Leg | No | Yes | Yes | Yes |
| Paul | 58 | 8 years | 2.5 years | Leg | Yes | Yes | Yes | Yes |

*The surgeon was an orthopaedic surgeon in most cases

**Pain clinic staff may include Pain Management Specialist, Psychologist, Psychiatrist, Nurse, Occupational Therapist, Physiotherapist, Exercise Physiologist and Dietician

health professionals each participant had seen during the course of their condition, demonstrating multiple encounters on which their opinion was based. The list is not exhaustive as participants were not asked to list each health professional they had seen.

## Data collection

Pilot interviews were conducted to refine the interview questions and ensure the objectives of the information gathering would answer the research question. During all interviews, a specific question was asked of each participant. "What do you think health professionals should know about your condition to treat you more effectively"? Data collected from pilot interviews, with participants that had been formally diagnosed with CRPS, were included in the analysis because the questions and interview format remained the same in the subsequent interviews.

Semi-structured interviews using the technique suggested by Seidman [25] were conducted by the first author in person or over Skype as per the participants' preferences and location. Interviews lasted between 30 and 90 minutes (*average = 51 minutes*) and were audio recorded and transcribed verbatim by either the first author or a professional transcriber.

The first author was diagnosed with CRPS in 2010 and was included in this research as a participant, being interviewed by another member of the research team following completion of the pilot interviews. This interview gave the first author insight into the experience of being a participant in this research leading to an empathetic approach to the interviews. The first author conducted all other interviews.

## Data analysis

The data were analysed manually by following the steps outlined by van Manen [16]. This included holistic, selective and detailed reading of the interview transcripts, identifying

fundamental phrases and asking questions of each line, sentence, and paragraph [16]. Additionally, the researchers applied the 'hermeneutic circle' which is the process of moving back and forth between data excerpts [26], studying sections of text and then looking at it as a whole [27], and the heuristic approach of taking a break and looking again [28].

**Rigour.** Rigour was ensured by following the suggestions of Johnston et al. [29] who discussed the inclusion of the researcher as a participant in a phenomenological study. Heuristics was included in the design to allow data from the lived experience of the first author, who has been diagnosed with CRPS, to be included as data without it dominating the research [28]. All participants chose a pseudonym to be used throughout the research except for the first author. The use of the author's name highlighted that her data did not dominate other participants voices and safeguarded against unintentional bias in the analysis. It also made explicit the preconceived understandings and assumptions of the first author which is necessary for phenomenological research [16]. Additionally, a reflexive journal was kept which ensured the decision trail could be followed by co-authors [27]. Essential and incidental themes in the data arose through the isolation of thematic statements and consensus agreements of the authors and the use of representative quotations from participants also adds transparency and trustworthiness to illustrate the interpretation of the findings presented [30]. In the writing of the findings, readers can then judge the credibility and possible generalisability, or transferability to their own settings [30].

## Findings

The overwhelming theme that emerged from the data was that health professionals do not know enough about CRPS. This lack of knowledge was thought by participants to lead to poor treatment decisions by clinicians which affected participants' health outcomes. The major theme, therefore, can be broken into five sub-themes which are:

- Don't touch unless I say it is okay

- Be patient with the patient/ It is important to develop a relationship

- Educate yourself and educate the patient

- Choose your words carefully

- Refer to others as needed

    An additional theme was also identified—It is very hard to describe CRPS.

    The major theme and sub-themes are now summarised followed by the second theme. This includes direct quotations from participants which serve as demonstration of how the findings emerged from the data [27].

### Theme A health professionals need to learn/know that CRPS exists

A common response is exemplified by Hannah who said, "Oh God, where to begin with that?. . . I think they should know more about CRPS. I don't think a lot of them do know about it." Carolyn also stated, "and the physios, because the physios don't understand either". All participants had encountered at least one health professional who had not heard of CRPS. While participants did not expect all health professionals to have extensive knowledge, some knowledge was reported to lead to a better understanding of symptoms and less wasted time in the initial stages of the disease.

*They wanted to get a second opinion and I had to wait at outpatients forever, and then I saw the resident, who had never heard of it, and then I had to wait for the registrar to come and he'd never heard of it. . . .. And then every time I went back to the hospital clinic, it was the same thing. I just had to sit and wait, and wait, and wait until the right person came. Colleen*

According to participants, lack of knowledge of CRPS (and chronic pain in general), often lead to disbelief by the health professional or the thought that symptoms were"psychosomatic". Those participants who had been injured at work expressed similar experiences with practitioners they dealt with through Work Cover.

*The pain specialist wanted permission to be able to put me on like a Ketamine infusion. Which wasn't given permission by Work Cover; they weren't going to cover that. So he then asked for a spinal cord stimulator, which wasn't approved either. Mel*

Many participants also assumed that had health professionals believed them they would have been diagnosed earlier. As stated by Dianne, "I'm fairly sure that if I had been diagnosed within that first year, within six months, three months, and had been using these treatments, that I would have made a full recovery". Many participants were unhappy with the initial care they had received when first experiencing symptoms and were aware that "if it's treated within in the first 3 months it can go into remission"

After commencing CRPS treatment, participants described difficulty getting specific advice. For example, Martin described an encounter between his wife and the local pharmacist when asking about medication side effects. Pain clinic staff did not always understand the changeable nature of CRPS, and one participant felt that some of the pain management specialists she had encountered "don't know about neurological pain" and said a neurologist had agreed with her.

*They should realise that each day is different. That pain is. . . you can't see a person's pain. A person on a good day might look perfectly normal. But that night they'd be very different. They need to realise that because a person can do one thing one day, or one week, doesn't mean they can keep doing it. The next week things might be very different. That pain and the limitations vary. Rosemary*

Jasmine expressed hope for better understanding of the physiological nature of CRPS leading to attitude changes in the future. She wanted health professionals to "realise how much actual pain we're in for a legitimate physiological reason. . . and until then it's not presumed that we're making it up". Such lack of knowledge and understanding was deemed unacceptable in some health professionals.

Within this major theme, there are five sub-themes which relate to specific topics that participants felt their health professionals should know regarding complex reginal pain syndrome.

**Sub-theme 1. Don't touch unless I say it is okay.** Besides knowing about the existence of the condition CRPS, participants felt that health professionals should know not to touch them without first asking permission. Alice found people touched her legs more often once she began using a wheelchair but stated, "I've seen them do it to other people in wheelchairs". This invasion of personal space heightened the participants feelings of discomfort and distrust. Touching the affected limb, particularly in the early stages of CRPS can be extremely painful. Doctors and nurses were considered the worst offenders of unexpectedly touching participants, even those clinicians with knowledge of CRPS

*I think that they definitely should know whether it's ok or not to touch you. Because I've even found like supposed pain specialists who will just reach out and grab you. Grab you on the leg or the foot without even saying is it ok to. And wondered why the hell you jump and just about put yourself through the ceiling. Karen*

Sharon's concern was the length of time it took her pain to settle after uninvited touch.

*If you walk past me the wrong way and you touch that arm, that's a big deal for weeks, months, hours if you're lucky. But people just can't grasp the concept of how much pain. I reckon that's the biggest hurdle. Sharon*

**Sub-theme 2. Be patient with the patient/ It is important to develop a relationship.**
Karen told of "Stumbling on the right doctor to find the right treatment" which was described by other participants as "going on a quest" and being "lucky". Having a rapport with a trusted health professional was considered necessary to managing CRPS. Participants taking Lyrica (pregabalin) described difficulty in elucidating clear thoughts which made consultations with new health professionals difficult as when clear answers could not be given to questions asked. Karen spoke of "feeling like an idiot" when being unable to coherently describe her pain to her specialist who did not know her well and said that at least her GP had known her for 27 years which made appointments with him "easier". Laura also found that her GP trusted her, accepted her words and felt he understood that how she looked did not always align with how she felt.

Lack of a shared history also affected those participants who had been hospitalised, particularly if the health professional did not attempt to build a rapport. One participant described being an inpatient and "some bloke that I've never met" making decisions about her without consultation with her, or her pain specialist. Just as this participant felt there had been assumptions made about her, so too did she make her own assumptions about the health professional. Her feelings at the time were, "They've got to do their pain rotation, which they hate, because. . .They're anesthetists because they don't want to talk to people. And they don't want people banging on about their pain".

Nurses were also mentioned but the care provided changed according to the ward the participant was admitted to. One participant described having a ketamine infusion on a surgical ward as "great because they are too busy", meaning that nurses did not have time to build close relationships and commence a discussion about their prescribed medication which often included opioid use. One participant stated that since health professionals have university degrees, they should know how "communicate" and "treat people as people", meaning, they should be building a rapport and providing care according to the individual needs of the patient.

Sharon felt judged by administration staff at a pain clinic when she told them about being unable to get to an appointment. She felt they "did not know her" or understand her mobility issues navigating the way from the taxi rank through a "huge hospital foyer with no seating" to the clinic.

*The one and only time I went on my own, they had said to me that there was a volunteer that would be able to help me to get around, and there wasn't. . . I just had to get up onto the next floor and I couldn't get there. I had to go home. Sharon*

Following such experiences, participants self-confidence was diminished, and they sometimes started to doubt themselves. Increasing their knowledge of CRPS though, helped the participants to reconcile their thoughts and rebuild their self-belief.

*And I go through it on a few occasions where I kind of go is there actually something wrong with me or am I just making this up? . . . I have a debate in my head for hours at a time. . . you can't fake the thermal imaging that was taken which quite clearly showed there's a three-degree difference in the temperature of my legs, that kind of thing you can't fake it. So there has to be something there. Martin*

**Sub-theme 3: Educate yourself and educate the patient.**   Participants who were also health professionals often had trouble finding advanced appropriate support because they were able to apply their disciplinary knowledge, but their body was sending different messages. They believed that the concepts involved in CRPS management are difficult to understand and unless a person lives through the experience, they never fully comprehend the condition. Two health professionals described allodynia when wearing a bra and shirt and one said,

*It took me a while to grasp that graded exposure in this neurological sense. It wasn't a cognitive thing; it was your neurological system. . . . there was no harm. It didn't matter what I thought. If that makes sense? But then again, I don't know if anybody with or without mental health experience could grasp that very easily anyway. It's quite a complicated thing to understand isn't it really? Dianne*

People are usually the experts in their own body and must become so when living with a long-term complex condition such as CRPS. A common thought was that if health professional students could meet and discuss CRPS with patients, they might better understand and believe future patients they encounter. Participants expressed willingness to help educate health professionals and many had similar ideas about their role in this education.

*They need to do a really good course and become more knowledgeable about this disease, and they need to talk to some people like me. They need to talk 1st hand to people who've had it. Rosemary*

Participants also recognised that self-management involves self-education, and that research is the key to a potential cure. Paul had participated in a research project, finding it a valuable educational experience.

*It's very good to talk about a patient's experience and see their different perspectives on things from a doctor's side of things, and from a patient's side of things. And sometimes those were two different views totally. Paul*

**Sub-theme 4. Choose your words carefully.**   The way a health practitioner engaged with the participants, impacted their self-beliefs concerning CRPS. Jackie, when diagnosed was told, "Yes, you have CRPS it's a very sinister complaint". The word sinister has remained with her, regularly reminding Jackie that CRPS is "really awful". Jasmine spoke of feeling "punished for something that's not my fault" when she felt that doctors were not responding to her appropriately. Jasmine also felt frustrated with her lack of improvement saying, "if your doctor doesn't even expect much of you then you just kind of resign yourself to this life of disability and isolation. And it's a lot more depressing than it needs to be". Verbal communication is an important component of building a therapeutic relationship between health professionals and patients. So too is nonverbal communication such as looking at the person and maintaining eye contact.

Alice felt that her wheelchair was the major obstacle in effective communication, not only with health professionals but the general community also. She felt people would defer to an "able-bodied" person if she were with one, and not direct questions or conversation to her and said that her disabled friend had reported similar experiences. Other participants described walking aids such as crutches being a distraction for people speaking to them and felt that the crutches were more a part of the conversation than they were. For people who were already feeling disbelieved and disrespected, this caused them to also feel as if they were unimportant and being ignored. Many participants voiced such feelings and Alice felt that the psychological problems that could be caused by CRPS needed to be acknowledged and treated along with the pain.

**Sub-theme 5. Refer to others as needed.** Participants felt it was the role of the doctor to refer them to other services as necessary and provided examples of "cutting edge treatment going on in Europe", "chronic pain support groups" and "websites" they had discovered through their own enquiries. Jasmine felt that doctors with little knowledge of CRPS were "so scared of a malpractice lawsuit that they just kick you out" instead of referring her to another health professional. She also described a negative experience after being referred.

*Because there was CRPS in that region, he told me that no matter what the imaging showed him, he would under no circumstances touch me. . .Any pain I have is just going to be related to the CRPS. It's almost like once that ICD9 code for CRPS; RSD gets somehow on your chart. Then all of your problems are somehow magically related to that, or you're magically untreatable because that's on your chart. . .It makes me feel like I'm being punished for something that's not my fault, Jasmine*

Many examples were given where a health professional did not acknowledge their limited knowledge of CRPS or complex pain issues. Participants appreciated those health professionals who were honest and admitted they did not have all the answers and felt that multidisciplinary teams were the most effective.

The second theme will now be described.

## Theme B. It is very hard to describe CRPS

Participants were aware that CRPS can present very differently from person to person. The ability of a person one day, could also be quite different the next. They found it difficult to accept that some health professionals did not understand this.

*What affects one person in one way, isn't going to affect a person in another way. . . . because my experience of pain, isn't going to be the same as someone who's got it in their arm, or their back, or wherever else you know. It's going to be a totally different sensation. Paul.*

Health professionals seemed to expect that participants could eloquently express themselves. Describing the pain at its worst while still being accepted as credible was particularly hard when speaking to health practitioners with no knowledge of CRPS.

*. . . but trying to convey the fact that this is the most ridiculous amount of pain, people can't fathom that. You're having me on. It can't be that bad. If it was that bad, I would have heard about this disease. Sharon*

One participant described taking video footage of herself to show to health professionals because she could not find the words to express the severity of her pain. The video was more readily accepted than her words indicating disbelief on the part of the health professional.

*It's so hard to explain that this pain is really severe. It's worse than childbirth. How do you explain it? I took a couple of movies of myself having a pain flare before the implant. That's when the pains were 10 out of 10 shooting pains. When I have those pains, they jolt my whole body. They come in succession. I'll be sitting there. . . every time I just. . . can't help but physically react to them. That's the closest I come to showing pain. And it really doesn't' tell anybody what it's like. Rosemary*

The consequences of an inability to adequately describe the pain to a health professional who had no understanding of CRPS were similar to the feelings described above. The participants felt disbelieved, disrespected and believed that if the health professionals knew more about CRPS, they would receive better care.

## Discussion

The major theme identified in this research is health professionals need to learn/know that CRPS exists. Participants felt that if their health professional knew more about the condition, they would not be disbelieved when they could not adequately describe their symptoms, pain levels and amount of suffering experienced. These issues are similar to those described by people living with other chronic pain conditions such as fibromyalgia [31]. The idea that pain does not exist once there is no evidence of it was first suggested around 1000 CE and unfortunately, such attitudes exist today and are exhibited in disbelief and accusations of malingering by health professionals who demonstrate a lack of empathy towards their patients [31].

The view of some health professionals that chronic pain patients are untrustworthy, depressed, drug-seeking or drug-abusing further increases the patient burden, compromises the patient/clinician relationship and results in poor levels of care [32]. Empathetic patient-centred care was found by Paul-Savoie et al. [33] to be more forthcoming when patients had visible physical signs of pain such as in rheumatoid arthritis than in conditions such as fibromyalgia. The physical signs of CRPS are known to reduce over time and some patients are loath to display or disclose their pain levels [18].

Participants expected that clinicians would respect their descriptions of the consequences of touching the affected limb. Therefore, uninvited touch led to distrust of the health professional concerned which compromised the therapeutic relationship. Therapeutic relationships are the relationships between a patient and a health professional and are focused on best patient outcomes, associated with patient satisfaction, quality of life, communication, trust, respect and patient centered care [34]. Within communication and patient-centred care, another foundational healthcare skill is providing information and gaining consent. Informed consent is a major ethical standard in research [35], is highlighted as a concern before surgery and is a means of providing permission for lawful touching [36]. Health professionals, particularly nurses, are educated about consent issues during their undergraduate degrees including the difference between expressed and implied consent such as the person positioning their body for their temperature to be taken [36]. It is surprising, therefore, that participants told of being touched on their affected limb without giving permission for it to be touched.

Aside from the consent issue, disbelief of the presence and intensity of their pain was offered by participants as another reason for clinicians touching a patient's affected limb without gaining prior consent. Health professionals working clinically are more likely than the general public to disbelieve people reporting chronic pain, and the higher the severity of the reported pain, the more likely it is to be discounted by clinicians. Nurses and physicians adjusting their empathy and practice according to the clinical condition as described by Paul-Savoie [33] may explain the lack of adequate care shown to some participants in this study. An

empathetic presence by the health professional is a factor in the delivery of patient-centred care which is related to positive outcomes for people living with chronic pain [33]. Additionally, patients perceive their treatment to be effective and show reduced symptoms and improved recovery when they feel their doctor has a good 'bedside manner' which is described as the approach of the doctor to the patient and their interaction style [37].

Development of an empathetic bedside manner and provision of patient-centred care rely on basic health care skills and effective communication. A therapeutic alliance is required where there is no power differential between the patient and the health professional, use of a biopsychosocial perspective and respect for the patient as a person to achieve effective patient-centred care [33]. Findings in this study support the notion that some health professionals lack adequate communication skills and do not provide effective patient-centred care. Back et al. [38] assert that most current health professionals, particularly physicians, have not undertaken adequate communication skills training because the basic principles of communication were not recognised up to 10 years ago.

Recent research indicates that pain education in general is lacking across all health curricula and students pain knowledge, beliefs and attitudes are not adequate, particularly for people living with long term chronic pain conditions [39]. Research suggests that limited pain content is taught in undergraduate health science courses [40, 41], and there is a discrepancy regarding the amount taught and the amount needed for effective practice [32]. There is no current stipulated amount of pain education nominated for undergraduate health students to cover within their degrees for registration as a health professional in Australia, although Hong Kong does have this requirement for nurses [42]. The International Association for the Study of Pain (IASP), the leading pain association in the world, has attempted to address this recognised shortfall.

IASP experts identified core concepts and developed nine curriculum outlines for pain education, suggesting these outlines be available free of charge, and be used throughout the world in medical and health professional education [43]. Pain education core curricula were first developed by the IASP in the 1990s and were further refined in 2017 to include an Interprofessional Pain Curriculum Outline in recognition that multidisciplinary pain care necessitates collaboration between the professions which requires comprehension of others roles and expertise [43]. Health students in Canada are reported to have improved pain knowledge as a result of using the IASP curriculum [44], however, it has been poorly integrated and pain education is still reported as inadequate in most countries [39]. Appropriate education of health professionals is important because current literature suggests that if acute pain is managed well when it occurs, the incidence of chronic pain is likely to be reduced [14]. This idea is also supported in a CRPS context [8].

Primary practitioners, such as GPs and physiotherapists were often the first to see participants in this research and, therefore have a large role to play in helping address this issue, a finding supported in the literature [8]. This notion is reflected in the European Taskforce Standard 15 "Physiotherapists and occupational therapists must have access to training in basic methods of pain rehabilitation and CRPS rehabilitation" [20]. The unique role played by pharmacists and medication management has also been shown to lead to better pain coping in patients, and to improved self-efficacy [45]. When assessing the cause of delayed CRPS diagnosis, Lunden and Jorum [19] suggest the cause is probably due to a lack of reporting and a lack of awareness. The authors in this current study suggest that all primary practitioners must have adequate education around the issue of chronic pain so that they can offer effective treatment for their patients. As reported by Boichat et al. [8], appropriate CRPS education of health professionals can lead to better patient education in self-management, and it is the responsibility of the health professional to provide high-quality resources.

In cases where the cause of the pain is unknown, it is also important for primary practitioners to consider CRPS as a possible diagnosis and to refer patients to other specialists, as necessary. The European Pain Federation list three standards for the diagnosis and management of complex regional pain syndrome that relate to referral [20]. The authors in this current study also suggest that implementation of these standards worldwide could result in better referral practices for all patients with chronic pain.

## Limitations

The number of participants was limited. This limitation was addressed through the use of in-depth interviews that allowed in-depth exploration with limited data collection points. Importantly we observed an agreement between participants in terms of major emerging themes. The non-inclusion of people who speak languages other than English must be recognised as a limitation in this research, however the interviewer only speaks English. Future studies with non-English speakers who suffer from CRPS should be conducted. The International Association for the Study of Pain advocates the use of patient voices in both research and education to achieve a humanistic, patient-centred approach to health care and to aid the understanding of the health care process and potential barriers to treatment effectiveness from the patients' perspective [46]. To further enhance research and practice people with other similar chronic pain conditions should be asked about their experiences to further investigate and inform this issue of health professionals' lack of knowledge. A better understanding of lived experiences of pain creates the potential to improve clinical pain management by enhancing comprehension and empathy from clinicians [47].

## Conclusion and recommendations

This paper reported on the answers people living with complex regional pain syndrome gave when asked the question, "What do you think health professionals should know about your condition to treat you more effectively"? The major theme that emerged from this research was health professionals do not know enough about CRPS. Within this theme, the following sub-themes were identified: don't touch unless I say it is okay; be patient with the patient/ it is important to develop a relationship; educate yourself and educate the patient; choose your words carefully, and, refer to others as needed. An additional theme of it is very hard to describe CRPS was also identified. The findings indicated that health professionals need to know of the existence of CRPS, require better knowledge about pain, and some, could improve their approach to patient care. Despite the IASP curriculum guidelines being developed and available free of charge, it would appear that little use is made of them by educational facilities. It is recommended that these guidelines be used to ensure a mandated minimum level of pain content is taught to all health students at the undergraduate level. Education about pain should be complemented by education in communication skills as this will lead to health professionals acquiring the skills necessary for an effective bedside manner resulting in better patient-centred care.

To the authors' knowledge, this is the first study examining the knowledge requirements of health professionals as determined by people living with CRPS. Recommendations arising from this study focus on primary practitioners. Participants felt that health professionals do not know enough about CRPS. Participants also wanted to be believed and treated appropriately. Therefore, the recommendations are increasing the knowledge of health professionals, improving their bedside manner and enhancing their capability to provide patient-centred care. This can be achieved through education on CRPS and pain in general and education on therapeutic communication of undergraduate and post-graduate health professionals.

According to Back et al. [38], communication skills training at the undergraduate level would take 10 years to result in a fully educated workforce, therefore a two-fold approach to education about pain and communication skills is suggested by the authors of this current study. Firstly, the inclusion of a unit on communication skills should be considered for all health students. Concepts such as therapeutic communication, consent, bedside manner, and acceptance of the patient voice are necessary. Health professionals should also have a mandated amount of pain-related content included in their undergraduate education. Such content needs to include the difference between acute and chronic pain and should be based on the IASP guidelines. Such education should also refer to CRPS because of the devastating consequences experienced by people who have this condition through delayed diagnoses and inappropriate treatment.

Despite the documented need for improved pain education in health students, most pain training is known to occur in the clinical setting [15]. Therefore, post-graduate, training in the clinical setting is also recommended. Focused workshops and on-site multidisciplinary professional development activities about pain, communication and patient-centred care will ensure that those health professionals who have not encountered this information at an undergraduate level, or those requiring further training, can gain the education necessary to provide effective evidence-based care.

## Supporting information

**S1 File. COREQ checklist.**
(DOCX)

## Acknowledgments

This research did not receive any specific grant from funding agencies in the public, commercial, or not-for-profit sectors. There are no conflicts of interest to declare.

## Author Contributions

**Conceptualization:** Colleen Johnston-Devin, Marion Gray.

**Data curation:** Colleen Johnston-Devin.

**Formal analysis:** Colleen Johnston-Devin, Florin Oprescu, Marianne Wallis, Marion Gray.

**Investigation:** Colleen Johnston-Devin.

**Methodology:** Colleen Johnston-Devin, Florin Oprescu, Marianne Wallis, Marion Gray.

**Project administration:** Colleen Johnston-Devin.

**Resources:** Colleen Johnston-Devin.

**Supervision:** Florin Oprescu, Marianne Wallis, Marion Gray.

**Validation:** Colleen Johnston-Devin, Marion Gray.

**Writing – original draft:** Colleen Johnston-Devin, Florin Oprescu, Marianne Wallis, Marion Gray.

**Writing – review & editing:** Colleen Johnston-Devin, Florin Oprescu, Marianne Wallis, Marion Gray.

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
