## [Decision Letter · Decision Letter 0]

24 Jan 2022

PONE-D-21-34889Educational considerations for health professionals to effectively work with clients with complex regional pain syndromePLOS ONE

Dear Dr. Johnston-Devin,

Thank you for submitting your manuscript to PLOS ONE. After careful consideration, we feel that it has merit but does not fully meet PLOS ONE’s publication criteria as it currently stands. Therefore, we invite you to submit a revised version of the manuscript that addresses the points raised during the review process.

Two Reviewers evaluated the manuscript giving generally positive opinions. However I agree with them that more details should be added about the usage of hermeneutic phenomenological approach and deeper interpretation of data. I also agree that the presentation of the entire poem is redundant, while Discussion should be more focused on actual findings of the study and relevant literature. Also are Authors sure there would be no issue with copyright by including an unattributed poem? What if the original author would claim it? Finally, Authors should add the COREQ checklist which is requested for qualitative studies on PLOS ONE. 

We look forward to receiving your revised manuscript.

Kind regards,

Stefano Triberti, Ph.D.

Academic Editor

PLOS ONE

Journal Requirements:

Reviewers' comments:

Reviewer's Responses to Questions

**Comments to the Author**

1. Is the manuscript technically sound, and do the data support the conclusions?

Reviewer #1: Yes

Reviewer #2: Partly

2. Has the statistical analysis been performed appropriately and rigorously? 

Reviewer #1: N/A

Reviewer #2: N/A

3. Have the authors made all data underlying the findings in their manuscript fully available?

Reviewer #1: Yes

Reviewer #2: Yes

4. Is the manuscript presented in an intelligible fashion and written in standard English?

Reviewer #1: Yes

Reviewer #2: Yes

5. Review Comments to the Author

Reviewer #1: Please see the attached document comments and respond accordingly. Did you include unpublished studies?

- Some descriptors were not used in the search and may have compromised and search strategy.

Exclusion criteria. The exclusion criteria are not clear. I didn't understand the first sentence

I suggest revision of English.

Reviewer #2: Manuscript title: “Educational considerations for health professionals to effectively work with clients with complex regional pain syndrome”

Overall comment

This manuscript presents a study which aimed to “To determine what people living with complex regional pain syndrome (CRPS) think health professionals should know about their condition to provide appropriate care”. It employed a heuristic, hermeneutic phenomenological study, and used semi-structured interviews with 17 participants living with CRPS.

The manuscript is well written, interesting, and relevant. I applaud the authors for using a qualitative approach to investigate the experiences of living with a syndrome that is very challenging for patients and health professionals.

There are some issues to consider that would, in my opinion, improve the manuscript. Here are some specific comments:

Background

The authors mention that “there is no global mandated content level of pain education for undergraduate health professionals` students…”. In my opinion, this sentence needs clarification since the International Association for the Study of Pian (IASP) has developed work in this area. I suggest a revision, as this may also ensure coherence between the information presented in the introduction and discussion sections.

Methods

Sample/ Participants

Information about participants (e.g. mean time lived with CRPS or mean time from symptom onset to formal diagnosis) is briefly presented and table 1 is indicated for more detailed data. I suggest the inclusion of a brief sentence about health professionals since it is also included in the table but not mention in the main text.

Data analysis

The link from reference (20) is not working.

Findings

Information presented in this section is quite interesting. Since the authors have used a hermeneutic phenomenological approach, I would expect a deeper interpretation of the data. On several occasions the manuscript is focused on merely describing what the participants had said and a quotation is presented. For example:

“Alice found people touched her legs more often once she began using a wheelchair but stated, “I’ve seen them do it to other people in wheelchairs”. Sharon’s concern was the length of time it took her pain to settle after uninvited touch.

If you walk past me the wrong way and you touch that arm, that’s a big deal for weeks, months, hours if you’re lucky. But people just can’t grasp the concept of how much pain. I reckon that’s the biggest hurdle. Sharon”

Additionally, considering the hermeneutic phenomenological approach I would not expect an idiographic focus on this analysis. The main text presented in the findings section calls attention to the individual`s participations, which would be expected in an interpretative phenomenological analysis (that integrates phenomenology, hermeneutic and idiography).

In my opinion, these two aspects may compromise the quality of the manuscript. I strongly recommend a revision of this section.

Discussion

The authors discussed the main findings of this study and introduced some references from relevant research.

I recommend the revision of two aspects (the introduction of a poem and the anecdotal evidence on veterinary science) explored in the next paragraphs.

I believe that the poem has provided a good inspiration and promoted reflection, which is very important for researchers working with qualitative data. However, in the discussion section the readers would expect a discussion focused on the relation between this study`s findings and previous research. I think the presentation of this poem does not add value for the discussion itself.

The authors mentioned that “anecdotal evidence exists supporting the notion that veterinary science undergraduate programs contain more pain content than that of a medical degree”. Since this is based on anecdotal evidence, I don`t think this information adds value to this discussion. The following sentence (“Research suggest that limited pain content is taught in undergraduate health sciences courses (5,31))” is clear, based on previous research and may be used without the anecdotal evidence.

Table 2. The authors could cite the reference. Is this table really needed?

Thanks

6. PLOS authors have the option to publish the peer review history of their article (what does this mean?). If published, this will include your full peer review and any attached files.

Reviewer #1: **Yes: **Helen Ali Ewune

Reviewer #2: **Yes: **Carmen Caeiro

---

## [Author Response · Author response to Decision Letter 0]

16 Mar 2022

Reviewer #1: Please see the attached document comments and respond accordingly. Did you include unpublished studies? 

 Thank you for your detailed suggestions to improve this manuscript. No unpublished studies were included. 

Some descriptors were not used in the search and may have compromised and search strategy.

 The initial literature search was rigorously undertaken. Updated articles have been included. page 3

Exclusion criteria. The exclusion criteria are not clear. I didn't understand the first sentence I suggest revision of English

 Participants did not have to prove their diagnosis. This is the same as previous research. page 6

Please include an introduction in your abstract.

 Introduction included as suggested page 2

The introduction looks like a problem. You need to rewrite it.

 Rewritten as suggested page 3

The flow or the coherence needs arrangement 

 Agree with the reviewer. The paragraphs have been rearranged to improve clarity. page 3

This should come first and should be introduced at the above 

 Thank you for this suggestion. Changed as suggested. page 3

you don't have to explain about this in the design section. All you have to mention is what design you followed 

 Sentence moved to Rigour section. “Heuristics was included in the design to allow data from the lived experience of the first author, who has been diagnosed with CRPS, to be included as data without it dominating the research”. page 8

How about the other language speakers? Do you think it is ethical to exclude the others as they have a lot to say about their condition.

 We agree that using only English speakers may be a limitation however as the interviewer does not speak any other languages inclusion of non-English speakers was beyond the scope of the research. Sentence about future research including non-English speakers has been included. page 22

Is it a data quality control or data collection procudure? you should make an arrangment to it 

 Data from pilot interviews were included in the analysis and therefore pilot interviews are included under the Data Collection heading. page 7

Was it manual or assisted by a software ? Not clear for me. 

 Sentence amended for clarity. The data were analysed manually by following the steps outlined by van Manen. page 8

How about the transferability issue and other rigour methods you haven't included and addressed them.

 The data analysis and rigour sections have been expanded to address this comment. page 8

You can put the themes in Thematic map rather than listing in a bullets 

 We attempted this but bullet points were clearer. page 9

It is not necessary to add quotes and 

 Explanation provided in Rigour section and commencement of Findings. pages 8, 9

We have to know who said this. and you better put it in a 

 This quote was from the first author who did not choose a pseudonym for transparency. page 10

Why do you mention the names

 Pseudonyms were used so readers could gain a sense of hearing the voice of a person, and identifying with them, rather than a number which evokes less emotional response. page 8

You are taking quotes as a theme but theme is a concept. you need to change it

 In this instance, the quotes provided the inspiration for naming of the theme. page 12

Very spliced topic... Please make it a theme

 The findings section has been expanded to address the review comments and a new theme created as suggested. page 21

You better make your discussion based on themes

 As per Reviewer 2, this has been revised and made clearer. page 9

?????

 Table 2 has been removed. 

The recommendation should come after your conclusion

 This has been moved as suggested. page 24

The limitation and your way to manage it should be clear

 Information has been added to this section to address this comment. pages 22, 23

Reviewer #2: Manuscript title: “Educational considerations for health professionals to effectively work with clients with complex regional pain syndrome”

Overall comment

This manuscript presents a study which aimed to “To determine what people living with complex regional pain syndrome (CRPS) think health professionals should know about their condition to provide appropriate care”. It employed a heuristic, hermeneutic phenomenological study, and used semi-structured interviews with 17 participants living with CRPS.

The manuscript is well written, interesting, and relevant. I applaud the authors for using a qualitative approach to investigate the experiences of living with a syndrome that is very challenging for patients and health professionals.

There are some issues to consider that would, in my opinion, improve the manuscript. Here are some specific comments: 

Thank you. We appreciate the guidance to improve the manuscript. 

Background

The authors mention that “there is no global mandated content level of pain education for undergraduate health professionals` students…”. In my opinion, this sentence needs clarification since the International Association for the Study of Pian (IASP) has developed work in this area. I suggest a revision, as this may also ensure coherence between the information presented in the introduction and discussion sections.

 Thank you for this suggestion. The information has been removed from the introduction. 

Methods

Sample/ Participants

Information about participants (e.g. mean time lived with CRPS or mean time from symptom onset to formal diagnosis) is briefly presented and table 1 is indicated for more detailed data. I suggest the inclusion of a brief sentence about health professionals since it is also included in the table but not mention in the main text. 

Added as suggested. pages 6, 7

Data analysis

The link from reference (20) is not working.

 The link has been fixed. page 8

Findings

Information presented in this section is quite interesting. Since the authors have used a hermeneutic phenomenological approach, I would expect a deeper interpretation of the data. On several occasions the manuscript is focused on merely describing what the participants had said and a quotation is presented. For example:

“Alice found people touched her legs more often once she began using a wheelchair but stated, “I’ve seen them do it to other people in wheelchairs”. Sharon’s concern was the length of time it took her pain to settle after uninvited touch.

If you walk past me the wrong way and you touch that arm, that’s a big deal for weeks, months, hours if you’re lucky. But people just can’t grasp the concept of how much pain. I reckon that’s the biggest hurdle. Sharon”

Additionally, considering the hermeneutic phenomenological approach I would not expect an idiographic focus on this analysis. The main text presented in the findings section calls attention to the individual`s participations, which would be expected in an interpretative phenomenological analysis (that integrates phenomenology, hermeneutic and idiography).

In my opinion, these two aspects may compromise the quality of the manuscript. I strongly recommend a revision of this section.

 Thank you, we agree with the reviewer. The findings section has been expanded to address the review comments. pages 9 - 22

Discussion

The authors discussed the main findings of this study and introduced some references from relevant research.

I recommend the revision of two aspects (the introduction of a poem and the anecdotal evidence on veterinary science) explored in the next paragraphs.

I believe that the poem has provided a good inspiration and promoted reflection, which is very important for researchers working with qualitative data. However, in the discussion section the readers would expect a discussion focused on the relation between this study`s findings and previous research. I think the presentation of this poem does not add value for the discussion itself.

The authors mentioned that “anecdotal evidence exists supporting the notion that veterinary science undergraduate programs contain more pain content than that of a medical degree”. Since this is based on anecdotal evidence, I don`t think this information adds value to this discussion. The following sentence (“Research suggest that limited pain content is taught in undergraduate health sciences courses (5,31))” is clear, based on previous research and may be used without the anecdotal evidence.

 Thank you for this suggestion. The poem has been removed. The anecdotal veterinary science information has also been removed. 

Table 2. The authors could cite the reference. Is this table really needed?

 Table removed as suggested and the reference cited. page 21

---

## [Decision Letter · Decision Letter 1]

2 May 2022

PONE-D-21-34889R1Educational considerations for health professionals to effectively work with clients with complex regional pain syndromePLOS ONE

Dear Dr. Johnston-Devin,

Thank you for submitting your manuscript to PLOS ONE. After careful consideration, we feel that it has merit but does not fully meet PLOS ONE’s publication criteria as it currently stands. Therefore, we invite you to submit a revised version of the manuscript that addresses the points raised during the review process.

The manuscript has improved. However I agree with Reviewer 2 that little effort has been done to modify findings section according to recommendations. Indeed the idiographic focus is missing. If Authors have used not IPA but a phenomenological approach that is supposed to have a nomotetic focus, this should be properly explained and justified. 

We look forward to receiving your revised manuscript.

Kind regards,

Stefano Triberti, Ph.D.

Academic Editor

PLOS ONE

Reviewers' comments:

Reviewer's Responses to Questions

**Comments to the Author**

1. If the authors have adequately addressed your comments raised in a previous round of review and you feel that this manuscript is now acceptable for publication, you may indicate that here to bypass the “Comments to the Author” section, enter your conflict of interest statement in the “Confidential to Editor” section, and submit your "Accept" recommendation.

Reviewer #1: All comments have been addressed

Reviewer #2: (No Response)

2. Is the manuscript technically sound, and do the data support the conclusions?

Reviewer #1: Yes

Reviewer #2: Partly

3. Has the statistical analysis been performed appropriately and rigorously? 

Reviewer #1: N/A

Reviewer #2: N/A

4. Have the authors made all data underlying the findings in their manuscript fully available?

Reviewer #1: Yes

Reviewer #2: Yes

5. Is the manuscript presented in an intelligible fashion and written in standard English?

Reviewer #1: Yes

Reviewer #2: Yes

6. Review Comments to the Author

Reviewer #1: (No Response)

Reviewer #2: Dear Authors

Thanks for the submission of a new version with ammendments. In my opinion, the changes on introduction and methods have improved the quality of the paper. However, the same has not happened in the findings and discussion sections. Although there has been an attempt to improve the depth of the interpretation, the problem related to the emphasis on the individuals` participation remains the same. As I mentioned before, I would expect an idiographic element in an interpretative phenomenological analysis. The authors have replied that they agree with this idea but have not made changes to address this aspect and have not present a justification to do so. In my opinion, this still compromises the quality of the paper. Additionally, the new version presents the findings together with discussion in the same section. This approach does not help the reader to follow the study and limits the depth of information presented in the findings and discussion sections. I would recomend the presentation of findings and discussion in different sections, as well as, the revision of findings in line with my previous comment (idiographic element).

7. PLOS authors have the option to publish the peer review history of their article (what does this mean?). If published, this will include your full peer review and any attached files.

Reviewer #1: **Yes: **Helen Ali Ewune

Reviewer #2: No

---

## [Author Response · Author response to Decision Letter 1]

9 May 2022

Thank you for your comment on the idiographic focus missing. You are correct. This study used phenomenology as described by van Manen (1990) who states, “phenomenology attempts to explicate the meanings as we live them in our everyday existence” (p.11). Our understanding is that nomothetic and idiographic perspectives are often used in psychology, but also used in ethnography and sociology. Nomothetic is usually quantitative and seeks to generalise. Although generalisation is not an aim of phenomenology (Crotty, 1996), it illuminates the essence of the phenomenon and the general structure of meaning which is a commonality through the experiences of each participant (van Wijngaarden et al., 2017). Idiographic approaches are usually qualitative and seek to explore what characteristics of a group separate it from another group. This was not the intent of this research and thus not discussed in this paper. 

Crotty, M. (1996). Phenomenology and nursing research. South Melbourne, Victoria: Churchill Livingstone.

van Manen, M. (1990). Researching lived experience: Human science for an action sensitive pedagogy. New York: State University of New York Press.

van Wijngaarden, E., Meide, H., & Dahlberg, K. (2017). Researching health care as a meaningful practice: Toward a nondualistic view on evidence for qualitative research. Qualitative Health Research, 27(11), 1738-1747. doi:10.1177/104973231771113

Comments to the Author

1. If the authors have adequately addressed your comments raised in a previous round of review and you feel that this manuscript is now acceptable for publication, you may indicate that here to bypass the “Comments to the Author” section, enter your conflict of interest statement in the “Confidential to Editor” section, and submit your "Accept" recommendation.

Reviewer #1: All comments have been addressed

Reviewer #2: (No Response)

6. Review Comments to the Author

Reviewer #1: (No Response)

Reviewer #2: 

Reviewer Comment Author Response

Dear Authors

Thanks for the submission of a new version with ammendments. In my opinion, the changes on introduction and methods have improved the quality of the paper. 

Response - Thank you

Reviewer - However, the same has not happened in the findings and discussion sections. Although there has been an attempt to improve the depth of the interpretation, the problem related to the emphasis on the individuals` participation remains the same. As I mentioned before, I would expect an idiographic element in an interpretative phenomenological analysis. The authors have replied that they agree with this idea but have not made changes to address this aspect and have not present a justification to do so. In my opinion, this still compromises the quality of the paper. 

Response - The findings section was expanded in line with the previous review comments. However, this research did not utilise IPA. This study used phenomenology as described by van Manen (1990) who states, “phenomenology attempts to explicate the meanings as we live them in our everyday existence” (p.11). Our understanding is that nomothetic and idiographic perspectives are often used in psychology, but also used in ethnography and sociology. Nomothetic is usually quantitative and seeks to generalise. Although generalisation is not an aim of phenomenology (Crotty, 1996), it illuminates the essence of the phenomenon and the general structure of meaning which is a commonality through the experiences of each participant (van Wijngaarden et al., 2017). Idiographic approaches are usually qualitative and seek to explore what characteristics of a group separate it from another group. This was outside the scope of this research and this paper. 

The Design section (Page 5) has been amended for clarity:

 “The hermeneutic phenomenological design chosen was informed by van Manen [16]. Acknowledging that the “(phenomenological) facts of lived experience are already meaningfully (hermeneutically) experienced” [16 p181], such research aims to describe, understand and interpret the experiences of the participants [17].This approach to phenomenology was chosen due to it using descriptions of lived experiences as data to describe the essence of a phenomenon [16]. This article reports on the data related to what health professionals should know about CRPS. The lived experience of living with CRPS is presented elsewhere [18]”.

Changes were previously also made in the Rigour section (Page 10) with the inclusion of the following two sentences:

“Essential and incidental themes in the data arose through the isolation of thematic statements and consensus agreements of the authors and the use of representative quotations from participants also adds transparency and trustworthiness to illustrate the interpretation of the findings presented (34). In the writing of the findings, readers can then judge the credibility and possible generalisability, or transferability to their own settings [34]”.

Reviewer - Additionally, the new version presents the findings together with discussion in the same section. This approach does not help the reader to follow the study and limits the depth of information presented in the findings and discussion sections. I would recomend the presentation of findings and discussion in different sections, as well as, the revision of findings in line with my previous comment (idiographic element). 

Response- The Findings and Discussion sections were combined because we misinterpreted the reviewers’ comment. These sections have again been separated.

---

## [Editor Report · Decision Letter 2]

19 May 2022

Educational considerations for health professionals to effectively work with clients with complex regional pain syndrome

PONE-D-21-34889R2

Dear Dr. Johnston-Devin,

We’re pleased to inform you that your manuscript has been judged scientifically suitable for publication and will be formally accepted for publication once it meets all outstanding technical requirements.

Kind regards,

Stefano Triberti, Ph.D.

Academic Editor

PLOS ONE
---

## [Editor Report · Acceptance letter]

3 Jun 2022

PONE-D-21-34889R2 

Educational considerations for health professionals to effectively work with clients with complex regional pain syndrome 

Dear Dr. Johnston-Devin:

I'm pleased to inform you that your manuscript has been deemed suitable for publication in PLOS ONE. Congratulations! Your manuscript is now with our production department. 

Kind regards, 

on behalf of

Dr. Stefano Triberti 

Academic Editor

PLOS ONE